# ATOMS AS WORDS: A NOVEL APPROACH TO DECIPHERING MATERIAL PROPERTIES USING NLP-INSPIRED MACHINE LEARNING ON CRYSTALLOGRAPHIC INFORMATION FILES (CIFs).

## ABSTRACT

In condensed matter physics and materials science, predicting material properties necessitates understanding intricate many-body interactions. Conventional methods such as density functional theory (DFT) and molecular dynamics (MD) often resort to simplifying approximations and are computationally expensive. Meanwhile, recent machine learning methods use handcrafted descriptors for material representation which sometimes neglect vital crystallographic information and are often limited to single property prediction or a sub-class of crystal structures. In this study, we pioneer an unsupervised strategy, drawing inspiration from Natural Language Processing (NLP), to harness the underutilized potential of Crystallographic Information Files (CIFs). We conceptualize atoms and atomic positions within a CIF similarly to words in textual content. Using a Word2Vec-inspired technique, we produce atomic embeddings that capture intricate atomic relationships. Our model, CIFSemantics, trained on the extensive Material Project dataset, adeptly predicts 15 distinct material properties from the CIFs. Its performance rivals specialized models, marking a significant step forward in material property predictions.

## 1  INTRODUCTION

In the field of condensed matter physics and materials science, predicting the properties of physical systems is an ubiquitous problem. Theoretically, a physical system can be described using a Hamiltonian based on the many-body interaction of the atoms in its lattice by considering microscopic degrees of freedom (Cubitt et al., 2018). Such a universal Hamiltonian can in principle describe all the properties of that system (las Cuevas & Cubitt, 2016). However, this task remains computationally impossible, with the current approach often combining approximations of the Schrödinger wavefunction equation and considering only a few body interactions between the atoms in the lattice and building different Hamiltonians to model different many-body phenomena (Luber, 2019; Diep et al., 2013). Some of the notable computational methods are density functional theory (DFT) or molecular dynamics (MD).

With AI's success in diverse fields (von Lilienfeld, 2020) and the availability of extensive open-access databases of material properties (Jain et al., 2011; 2013; Kirklin et al., 2015), researchers are leveraging ML techniques to predict material properties from input features (Isayev et al., 2017; Das et al., 2023; Zhuo et al., 2018). The primary hurdle, however, lies in aptly representing materials in feature space as a fixed-length vector, which is compatible with most ML routines (Wu et al., 2020; Himanen et al., 2020). To address this, attempts have been made to engineer feature vectors or descriptors using handcrafted elemental or calculated properties of constituent atoms, typically related to their periodic table positions, electronic structures, and other physical attributes (Ward et al., 2016; Meredig et al., 2014). However, such descriptors are crystal structure agnostic and can only be applied to small subset of crystal structures (Li et al., 2021). Recent efforts have used graph neural networks to represent the crystal structure by a crystal graph (Xie & Grossman, 2018; Gurunathan et al., 2023), encoding both atomic information and bonding interaction. This approach successfully captures the local chemical semantics around the atom but

fails to encapsulate important global periodic structural information (Das et al., 2023). Therefore, while existing methods have made progress in material representation, there are still gaps in capturing all the necessary properties effectively.

In this work, inspired by the success of NLP in understanding the meaning of words and sentences (Mikolov et al., 2013), we propose a unique approach to learn a universal representation of a material structure, atoms, and interactions with the capacity of predicting as many properties as needed. Traditionally, in material property calculations through computational methods, structural details are provided via a Crystallographic Information File (CIF). This text-like document lists atoms and their XYZ atomic positions within the unit cell. Viewing it through the lens of NLP, this situation mirrors how words are contextualized in a sentence. Just as words derive meaning from their neighboring words, atoms in a CIF can be understood based on their proximate atomic neighbors and atomic positions. Expanding on this concept, when analyzing numerous CIF files across diverse materials, the model begins to discern patterns: recognizing atom similarities, frequently paired atoms, and understanding inherent three-dimensional spatial relationships between them. Moreover, given that all atoms and their positions in the unit cell are explicitly mentioned, insights into the inherent crystallography symmetry, along with local and global structural information, are naturally ingrained. Furthermore, drawing parallels with NLP, where downstream tasks such as sentiment analysis and language translation deepen the understanding of a language, predicting a spectrum of material properties could help the model to learn richer embeddings of the atoms and atomic positions. In this study, we highlight the untapped potential of CIFs, traditionally employed for representing crystal structures in computational tasks and crystallography, as a rich source for machines to learn about atomic properties and crystal structures in an unsupervised manner. This represents a novel direction in the field. Our model, CIFSemantics, has been benchmarked for predicting multiple properties simultaneously. It demonstrated the ability to concurrently predict 15 distinct properties, achieving accuracy metrics comparable to SOTA models that are heavily reliant on handcrafted features designed for individual target properties or designated compound groups. We found that vector embeddings of all the atoms naturally showed similarities to patterns in the periodic table, reflecting each atom's intrinsic nature. We further demonstrated the robustness and adaptability of the learned atomic embeddings in three different prediction tasks where only the stoichiometry is known, emphasizing its potential in scenarios where detailed structure is unavailable.

## 2 METHODOLOGY

### 2.1 DATA SET

The crystallographic information and properties of all the available materials were downloaded from the Material Project database via its API. The dataset contained about 150,000 entries of inorganic materials. For each material entry, we retrieved the following 15 properties that were available for all the entries: volume, density, density_atomic, uncorrected_energy_per_atom, energy_per_atom, formation_energy_per_atom, energy_above_hull, is_stable, band_gap, efermi, is_gap_direct, is_metal, is_magnetic, ordering, total_magnetization_normalized_formula_units in addition to crystallographic information. Out of these 15 properties, 5 of them are categorical features whose distribution is shown in Figure 2. Notably, only one-third of the materials are stable. The retrieved crystallographic information had to be pre-processed to create text files for each compound. We included only the information that is typically available in the CIF file Hall et al. (1991): compound name, lattice constant, angles, space group, point group, atoms in the unit cell, and atomic positions. We created a cleaner, simplified format for writing the crystallographic information (as shown in Figure1) which we will henceforth refer to as the CIF file. In our version, we listed all the atoms in the unit cell, and the number of lines in these text files varies depending on the number of atoms in the unit cell.

### 2.2 TOKENIZATION

Given that our work entailed word-to-vector embedding, each unique, space-separated word was assigned a unique token. Since our goal was for the model to infer the properties of a compound from its constituent atoms and their positions, we splitted each compound name into its unique elements and their stoichiometry. For instance, the compound Ho1Bi1 was represented as Ho 1 Bi 1. Tokenizing numerical data such as lattice constants, atomic positions, and stoichiometry numbers posed

a significant challenge in our study. While numbers are usually filtered in word2vec applications Trewartha et al. (2022) due to their infinite variations, tokenizing these numbers was crucial in our context. Addressing this, we devised a method where we rounded off numbers to limit the range of unique values and appended an identifier string to differentiate between various categories. For instance, lattice angles - varying between 0 and 180 degrees - were discretized into integer values, each suffixed with '_deg'. This allowed us to differentiate these from other data types, such as atomic positions. Atomic positions, lying between 0 and 1, were discretized in steps of 0.01 and appended with '_ap'. Through this approach, we allocated a unique token for each numerical words.

### 2.3 PRE-TRAINING

To learn the embedding of each word token in the CIF documents, the Skip-gram method was employed. This method, which operates in an unsupervised fashion, aims to optimize word embeddings by maximizing the probability of observing surrounding context words $W_{t+j}$ for a given central word $W_t$ within a specified context window of size $c$:

$$\max \frac{1}{T} \sum_{t=1}^{T} \sum_{-c \leq j \leq c, j \neq 0} \log P(W_{t+j}|W_t) \tag{1}$$

With a large corpus encompassing approximately 24 million words, our refined vocabulary was limited to just 3,385 unique words. This provided a sufficiently dense sampling to effectively discern the underlying semantic relationships.

### 2.4 PREDICTION PROPERTIES PRE-PROCESSING

After learning the initial embedding from the semantic relationships of words in CIF files that described materials, we refined these embeddings by predicting the 15 properties previously mentioned. In preparing the data for the prediction model, scaling the properties was essential to ensure each property's equal impact on the prediction error. We observed that some properties, whose values spanned multiple orders of magnitude, displayed significant skewness. To address this, we applied a logarithmic +1 scaling for properties such as volume, density, formation energy per atom, and total magnetization. For other properties, like energy per atom and Fermi energy, standard scaling was employed to remove the mean and scale to unit variance. The energy above hull, with approximately 30% of its samples at 0 eV and the remainder ranging from 0 to 10 eV, underwent a power transformation. Full details about each property, along with their original and adjusted distributions, can be found in Figure 6. For categorical features such as ordering, stability, and is_metal, we used integer labels. This approach allowed us to use the same model for both categorical and continuous targets. By treating the encoded categorical values in the same regression framework as continuous properties, we ensured uniform handling of all properties within the prediction model. We also took precautions to prevent any information leakage from the training to the test set during distribution normalization. After training, we evaluated the model's accuracy on categorical variables by rounding the output to the nearest integer and comparing it with the integer target.

### 2.5 FINE TUNING

To fine tune the embeddings using regression task, we employed an LSTM (Long Short-Term Memory) network (figure 1) where the tokenised crystallographic file was the input, and the output was linked to a prediction layer. The model's weights, including the embeddings, were optimized using a combined RMSE loss calculated from the predictions of all 15 properties:

$$\text{Loss} = \sqrt{\frac{1}{15} \sum_{i=1}^{15} (\text{predicted}_i - \text{actual}_i)^2} \tag{2}$$

where $\text{predicted}_i$ is the predicted value of the $i^{th}$ property and $\text{actual}_i$ is its true value. This resulted in a model that could ingest a crystallographic file and predict its properties, bridging the gap between crystallographic text data and property prediction.

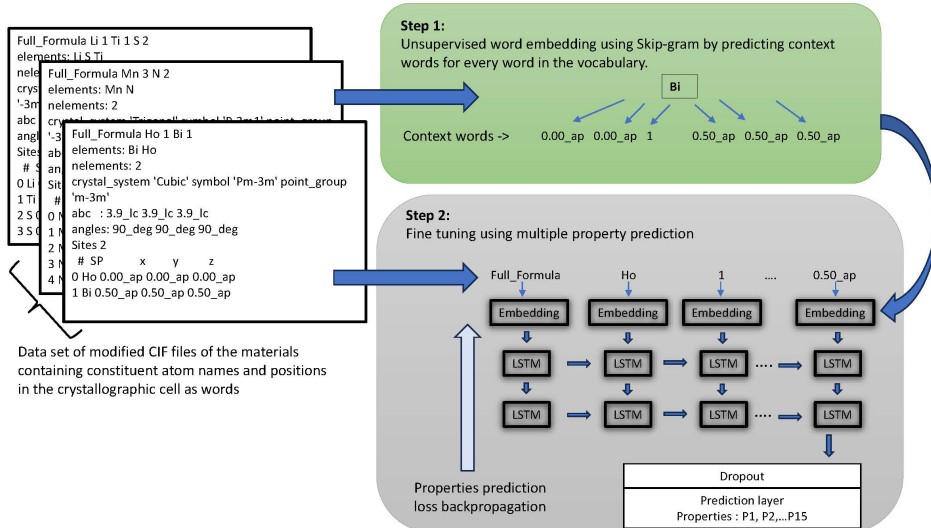

Figure 1: Workflow and model. CIFs, which contain crystallographic information such as constituent atoms and atomic positions, are transformed into a vector representation using the Skip-gram word2vec technique. The resultant word embeddings are then input to an LSTM architecture, with the final layer dedicated to predicting 15 properties. The prediction loss is backpropagated to optimize the model's weights, including the embeddings themselves.

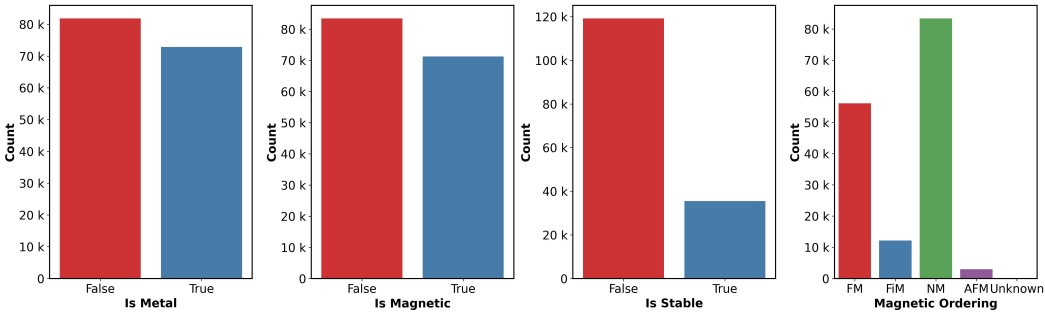

Figure 2: Distribution of the counts of metallicity, magnetism, stability, and type of magnetic ordering for the compounds obtained from the Materials Project database. Notably, only less than 40k, or one-third, of these compounds are stable, revealing a significant imbalance in the class.

## 3 EXPERIMENT

During the pre-training phase, we adopted the Skip-gram method implemented in the Python library Gensim (Řehřek et al., 2011), to generate embeddings across various context window sizes: 20, 40, 60, and 80, coupled with embedding vector lengths of 50 and 100. Each of these embedding variants was then utilized as the initial input word embeddings for the LSTM model. The LSTM architecture was implemented using the Pytorch library. For the LSTM configuration, we explored hidden layer sizes of 100 and 200, LSTM layer counts ranging between 2 and 4, and dropout rates of 0.1 and 0.2. The data was divided into a 70-30 train-test split. We found the best model used an embedding vector dimension of 100, a context window length of 20, hidden layer of size 200, 4 LSTM layers, and a dropout rate of 0.1.

## 4 RESULTS

In this section, we will look at the learned embeddings and the performance of our best model in predicting different properties in detail. To assess whether the learned embeddings for the atoms encode meaningful materials science knowledge, we projected the 100-dimensional embedding vectors of the atoms onto two dimensions using t-SNE (Van der Maaten & Hinton, 2008). Notably, these embeddings, as shown in Figure 3, closely resembled the layout of the periodic table, indicating that the embedding captured fundamental chemical characteristics. After the initial pre-training with Skip-Gram, the atom embeddings already formed clusters that are reminiscent of periodic table groups, such as alkali metals, lanthanides, and transition metals (Figure 3 (a)). Upon fine-tuning with property prediction tasks using LSTM, these clustered groups became more distinct. Additionally, within a cluster, atom begin to arrange in a manner that reflects their sequence within rows or columns of the periodic table. It's crucial to note that no explicit chemical information was provided for any of the elements – they were simply represented as strings. The model's understanding of each atom was solely derived from its position within the CIF file and the subsequent prediction tasks.

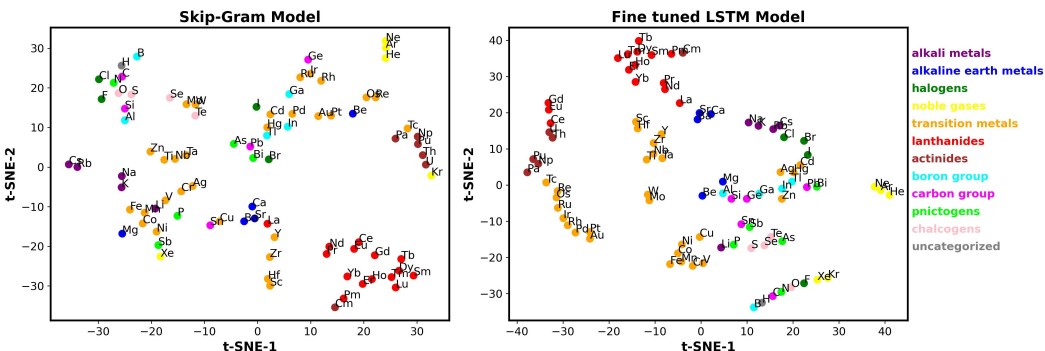

Figure 3: 2D t-SNE visualization of 100 dimensional word embeddings for elements. Embeddings are derived from crystallographic text files using the Skip-Gram model's masking technique (a) and subsequently fine-tuned via an LSTM model predicting 15 targets simultaneously (b). The embeddings exhibit clustering patterns that resonate with the periodic table's organization, highlighted through color coding. This shows embeddings have intrinsically grasped similarities and differences among elements based solely on their contextual occurrences in the text files.

Next, we look at the model's prediction capabilities on 30,000 samples in the test set. As mentioned earlier, the model predicts 15 properties simultaneously. Figure 4 presents the parity plots comparing predicted and true values for selected properties: Fermi energy, Energy per atom, Formation energy per atom, and Bandgap. A detailed performance metric for all the regression properties is tabulated in Table 1. In all regression tasks, except energy above hull ($E_{hull}$), the model achieved $R^2 > 0.7$ with four properties even surpassing 0.95. The lower $R^2$ for $E_{hull}$ is attributed to the distribution of $E_{hull}$, which is zero for all stable compounds (representing one-third of the dataset) and varies between 0 and 10 for the remaining compounds. For such cases, the Mean Absolute Error (MAE) is more indicative, registering an impressive 0.09 eV MAE.

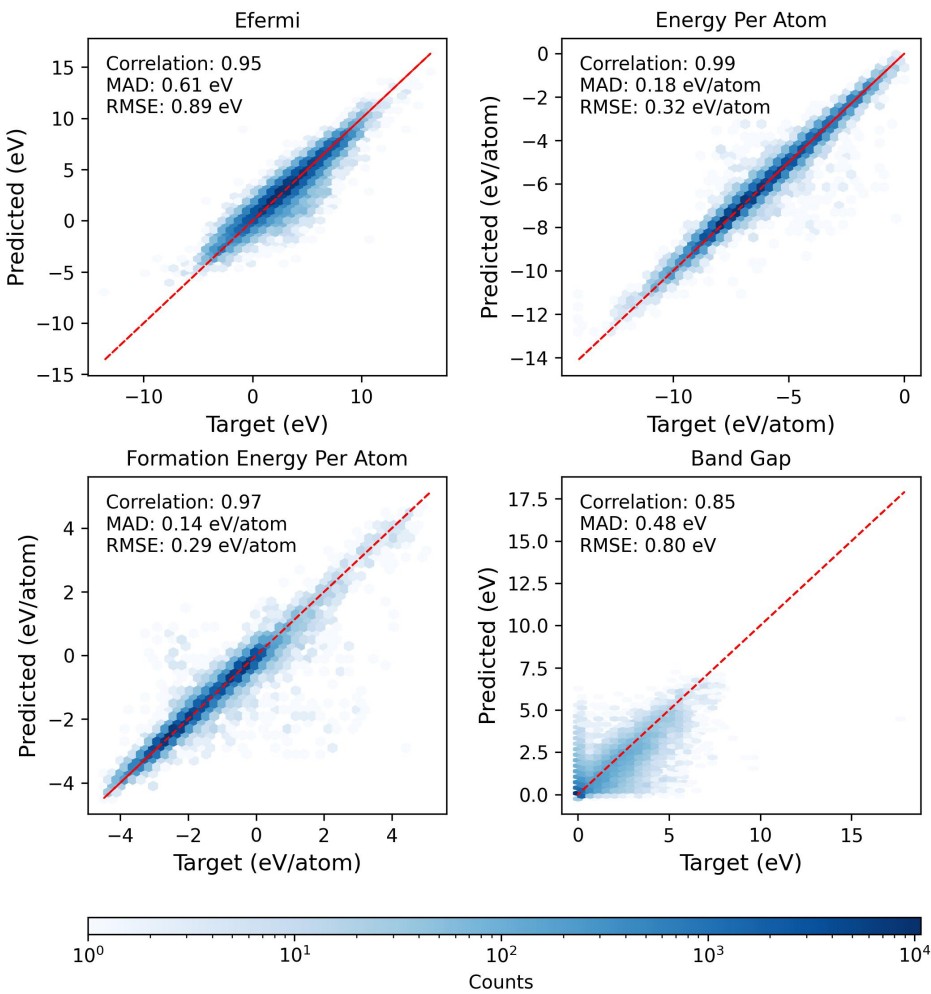

Figure 4: Performance of the LSTM model on the test set for four representative properties. The red dashed line denotes perfect prediction, while the color intensity indicates the density of data points. The model robustly predicts multiple properties simultaneously with high accuracy.

Table 1: Model Evaluation Metrics

| Feature Name | MAD | MAE | RMSE | $R^2$ |
|---|---|---|---|---|
| volume | 46.58 | 41.70 | 210.88 | 0.885 |
| density atomic | 2.158 | 1.933 | 16.80 | 0.844 |
| band gap | 0.457 | 0.451 | 0.789 | 0.721 |
| formation energy per atom | 0.140 | 0.139 | 0.269 | 0.950 |
| energy per atom | 0.170 | 0.173 | 0.297 | 0.975 |
| energy above hull | 0.115 | 0.091 | 0.311 | 0.391 |
| total magnetization normalized | 1.351 | 1.160 | 3.222 | 0.781 |
| uncorrected energy per atom | 0.171 | 0.173 | 0.295 | 0.972 |
| efermi | 0.622 | 0.622 | 0.910 | 0.893 |
| density | 0.399 | 0.400 | 0.595 | 0.954 |

In the forthcoming analysis, we benchmark our model's performance with other models in the literature. Many studies have focused primarily on predicting the band gap. Our model achieved a MAE of 0.451 eV. For reference, Xie & Grossman (2018) using graph convolutional neural and Li et al. (2021) employing elemental descriptors, reported MAEs of 0.388 eV and 0.341 eV respectively for

perovskite materials. In contrast, Sanyal et al. (2018) achieved an MAE of 0.295 eV. and Espinosa et al. (2022)'s 3D orthogonal vision-based approach yielded an MAE of 0.678 eV, while Sayeed's study Sayeed et al., leveraging word embeddings for crystal structure description, achieved an MAE of 0.475 eV. Shifting focus to the energy per atom ($E_{atom}$), our model achieved an MAE of 0.17 eV, outperforming Qu et al. (2023), who incorporated MatSciBERT and MatBERT embeddings, and achieved 0.29 eV. Regarding Fermi energy predictions, our model, with an RMSE of 0.9 eV, clearly excels over the DOS transformer method by Lee et al. (2023) which noted an RMSE of 1.4 eV. It's worth mentioning that several of the mentioned studies either relied on specific features tailored to particular compounds or mainly focused on a single property prediction.

Next we look at the prediction capability for the categorical properties of inorganic compounds, namely whether they are metals, stable, have a direct band gap, are magnetic, and type of magnetic ordering. The model's continuous outputs were rounded to the nearest integer and compared to the discrete integer class labels. The accuracy and F1 score are tabulated in Table 2.

Table 2: Performance on categorical features

| Property | Accuracy (%) | F1 Score |
|---|---|---|
| is_metal | 85.78 | 0.86 |
| is_stable | 84.18 | 0.76 |
| is_gap_direct | 87.56 | 0.69 |
| is_magnetic | 91.91 | 0.92 |
| Magnetic Ordering type | 82.42 | 0.60 |

Our model attained an accuracy of 86% for the metal/non-metal classification task. This is nearly comparable to the 92% accuracy achieved by Zhuo et al. (2018) on a more limited dataset of 1,000 compounds using compositional descriptors. It also surpasses the 76% accuracy of Zhou et al. (2018)'s Atom2Vec model. For the nature of band gap prediction task, our model achieved an accuracy of 87.5%, which is close to the 91% accuracy achieved by Mattur et al. (2022) who focused only on perovskite oxides. Our model also achieved a high F1 score of 0.76 for the stability prediction task, despite the large class imbalance in the dataset. Finally, our model achieved an accuracy of 91.9% for the magnetic compound classification task.

In our final evaluation step, we focused on the potential of the atomic word embeddings directly, without leaning on the full model which required CIF text data. This is relevant for many cases where only the chemical formula or the atoms in a material are known, and not the detailed structure. Furthermore, we aim to demonstrate that our atom embeddings, while trained for specific tasks, can also be applied effectively in contexts beyond their original design, showing their wider utility. We picked three tasks for the demonstration: predicting the Curie temperature for face-centered ferromagnets, estimating the solute diffusion barrier in metals, and screening fractionally doped perovskite oxides (FDPO). Predicting Curie temperature of ferromagnets is a complex task and is often modeled using Heisenberg-type interaction and Monte Carlo sampling Nolting et al. (1995). It's interesting to consider if our embeddings, trained mainly from patterns in the CIF files can be used to capture this complex interaction. Using the dataset from Belot et al. (2023), where they worked with a stoichiometry-based one-hot encoding for atoms, we applied our embeddings. We took a weighted average of the constituent atom's word vectors based on stoichiometry to get fixed-length vectors, as shown in Figure 5(a) as input to XGBoost model. Our method achieved an MAE score of 70.52 K, which is close to the 71 K reported by them. On the other hand, contrary to prediction properties considered so far which are specific to interactions originating from the atoms within a crystal structure, diffusion is a phenomenon where atomic migration and kinetics are involved. This is a significant shift from the previous tasks our embeddings were trained on. For this, we used the same dataset as Wu et al. (2017). We concatenated the embeddings of the host atom with that of the impurity atom and used this as input for XGBoost as shown in figure 5 (b). This approach gave us an RMSE of 0.21 eV, which is quite close to the 0.15 eV they achieved using several hand-crafted features. Lastly, the third task highlights how our embeddings can effectively represent disordered systems, a task that traditionally needs considerable effort (Torrisi et al., 2023). FDPO materials, as discussed by Zhai et al. (2022), are versatile and have many applications like energy conversion, storage, catalysis, and superconductivity. We decided to test our method using the same data Zhai et al. (2022) used to predict stable FDPO compositions. To achieve this, a fixed-

length feature was formed by combining the embeddings of each atom in the compound with its stoichiometry. The input format was [embedding of atom 1, stoichiometry 1, embedding of atom 2, stoichiometry 2, ...]. For formulas with fewer atoms, padding was added to ensure a consistent embedding vector length. A more detailed schematic of the flow can be found in Figure 5 (c). Using the RandomForestClassifier for predictions, an accuracy of 95.7% was attained, closely matching the 95.4% reported by Zhai et al. (2022).

Table 3: Comparison of Our Model's Performance with Other Studies

| Property/Task | Metric | Our Model's Score | Comparison Studies & Scores |
|---|---|---|---|
| **LSTM Model: Refining Embeddings + Multiple Property Prediction** | | | |
| Band gap | MAE (eV) | 0.451 | Espinosa et al. (2022): 0.678 Sayeed et al.: 0.475 Xie & Grossman (2018): 0.388 Li et al. (2021): 0.341 Sanyal et al. (2018): 0.295 |
| Energy per atom | MAE (eV) | 0.17 | Qu et al. (2023): 0.29 |
| Formation energy per atom | MAE (eV) | 0.139 | Zhou et al. (2018): 0.27 |
| Fermi energy | RMSE (eV) | 0.9 | Lee et al. (2023): 1.4 |
| Metal/Non-metal classification | Accuracy | 86% | Zhou et al. (2018): 76% Zhuo et al. (2018): 92%; |
| Nature of band gap | Accuracy | 87.5% | Mattur et al. (2022): 91% |
| **Single Property Prediction Using Learned Embedding** | | | |
| Curie temperature | MAE (K) | 70.52 | Belot et al. (2023): 71 |
| Solute diffusion barrier | RMSE (eV) | 0.21 | Wu et al. (2017): 0.15 |
| Stable FDPO compositions | Accuracy | 95.7% | Zhai et al. (2022): 95.4% |

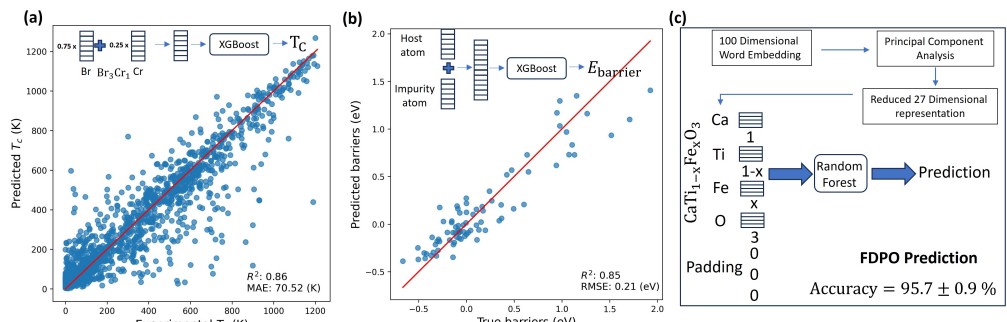

Figure 5: Efficacy of learned atomic word embeddings for three diverse tasks: Curie temperature prediction in ferromagnets, activation energy determination for dilute impurity diffusion in metals, and screening of fractionally doped perovskite oxides (FDPO). (a) Word embeddings of the constituent atoms are averaged according to their stoichiometric ratio, subsequently serving as the input feature to the XGBoost model for Curie temperature estimation. (b) The embedding vectors of the host and impurity atoms are concatenated to generate the feature vector. For both tasks, the model showcased a high $R^2$ value. (c) The 100-dimensional word embeddings were initially reduced to 27 dimensions using PCA, capturing 95% of the variance. Each reduced embedding of a compound's constituent atom was concatenated with its stoichiometry value. To ensure a consistent input length, zero-padding was added as needed, with the final feature vector serving as input to the Random Forest classifier, achieving an accuracy of 95.7% on stratified cross-validation.

## 5 CONCLUSION

In this study, we presented a novel approach to material representation inspired by Natural Language Processing (NLP), emphasizing the untapped potential of Crystallographic Information Files (CIFs)

as a robust knowledge base for machine interpretation of materials. By interpreting atoms and their positions in crystallographic text files (CIFs) as words in a sentence, we have demonstrated that our method can derive chemically meaningful embeddings for atoms. These embeddings naturally align with observed patterns and clusters in the periodic table. Furthermore, we have used this approach to simultaneously predict multiple properties with an accuracy comparable to state-of-the-art models which often excel in narrower classes of materials and are often tailored for single-property predictions. We anticipate that as we diversify our range of prediction tasks, each reflecting the material at distinct energy and time scales, there's a potential for our model to continuously refine its understanding of the inherent laws of physics and learn the many-body interaction. Furthermore, our framework is versatile. In cases where detailed structural data is lacking, the learned embeddings can represent the stoichiometry-based composition of materials directly. This adaptability is further exemplified in our successful predictions of properties like curie temperature, diffusion energy barrier, and FDPO screening.

Moving ahead, while our framework has already demonstrated its potential in various tasks, it also highlights avenues still to explore. One primary improvement lies in refining the representation of numerical tokens such as atomic positions in CIF files beyond mere binning. Exploring transformer architectures might be worthwhile, given their effectiveness in understanding context through attention mechanisms. Moreover, integrating prediction tasks, like diffraction patterns and Pair Distribution Function (PDF), could train a model to learn the average and local structure of material respectively.

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

## A    APPENDIX

You may include other additional sections here.

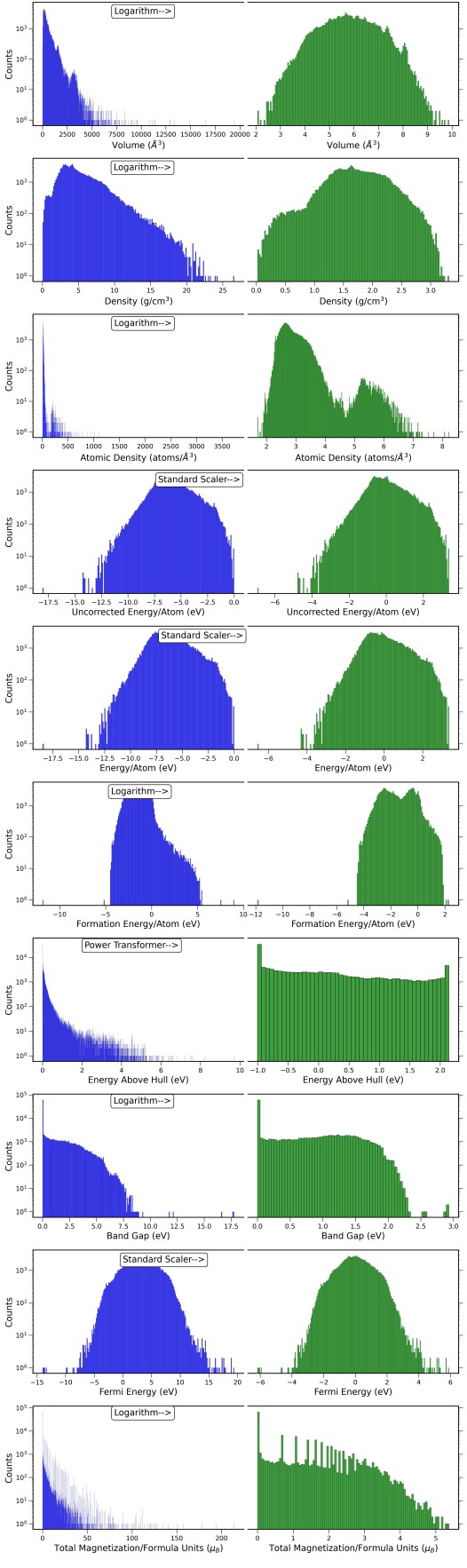

Figure 6: Distribution of different properties before and after transformation.

