# OpenReview forum: "Atoms as Words: A Novel Approach to Deciphering Material Properties using NLP-inspired Machine Learning on Crystallographic Information Files (CIFs)"
_ICLR.cc/2024/Conference — ICLR 2024 Conference Withdrawn Submission_

### Official Review · Reviewer_2xVF · 2023-10-29

**Soundness:** 3 good
**Presentation:** 2 fair
**Contribution:** 2 fair
**Rating:** 3
**Confidence:** 5

**Summary:**

This paper explores a natural language processing (NLP)-based approach to predict material properties from crystal information file (CIF) documents, which encode crystal structures in text format. The authors employ a skip-gram approach to learn word embeddings for individual tokens from the CIFs, and then use an LSTM network to fine-tune the learned embeddings to predict 15 distinct material properties. Specialized tokenization methods are developed to encode lattice parameters, fractional coordinates, and other features. Empirically, the performance of the property prediction approach is generally worse than most graph neural network (GNN)-based baselines, but it achieves comparable performance on some tasks such as band gap prediction.

**Strengths:**

The success of large language models like GPT-4 and LLaMA 2 has led to an exploration of alternative methods for predicting material properties, such as using the text representation of the material. This paper explores this new direction and compares their approach to classical GNN-based methods.

Specialized tokenization approaches are developed for the CIF files to encode float numbers like lattice angles and fractional coordinates. These approaches are not completely novel, as CrystaLLM [1] also develops some specialized tokens for CIFs.


[1] Antunes, Luis M., Keith T. Butler, and Ricardo Grau-Crespo. "Crystal Structure Generation with Autoregressive Large Language Modeling." arXiv preprint arXiv:2307.04340 (2023).

**Weaknesses:**

The results reported in this paper are substantially worse than those of simple GNN baselines for most properties, especially for representative properties such as formation energy and band gap. Since CIF files contain the full information of crystal structure, it is straightforward to convert this information to a graph representation and then predict the properties using a GNN. The substantially worse performance indicates that the approach is not going to be practically useful.

In addition, the formation energy prediction performance is even worse than that of a composition-only model like CrabNet in MatBench [1, 2], indicating that the model fails to learn some simple correlations between properties and elements.

The benchmark comparisons are not well structured. The authors use different baselines for different properties in Table 3, which makes it difficult to draw clear conclusions because each baseline is trained using different datasets and settings. Benchmarking on standard datasets like MatBench [2] would provide more useful statistics for the community.

[1] https://matbench.materialsproject.org/Leaderboards%20Per-Task/matbench_v0.1_matbench_mp_e_form/
[2] Dunn, Alexander, et al. "Benchmarking materials property prediction methods: the Matbench test set and Automatminer reference algorithm." npj Computational Materials 6.1 (2020): 138.

**Questions:**

I suggest that the authors report the performance of their method on MatBench for a fair comparison with other approaches. Given the current results reported in Table 3, I am not convinced that this approach is practically useful, as it performs substantially worse than simple GNN baselines. I am open to changing my mind if the authors report significantly better performance on MatBench.

---

### Official Review · Reviewer_NdwN · 2023-10-29

**Soundness:** 2 fair
**Presentation:** 1 poor
**Contribution:** 2 fair
**Rating:** 3
**Confidence:** 4

**Summary:**

The paper proposes a technique for representation learning on Crystallographic Information Files (CIFs) inspired by word2vec called CIFSemantics. The paper first introduces and motivates the challenge of materials property prediction using machine learning and the proceeds to describe the potential application of NLP techniques to learning material representations based on CIF files, which are text-like documents.  The introduction describes details related to learning the embeddings themselves as well as how downstream tasks can be useful in that process.

Next, the paper describes the general methodology, including the dataset (Materials Project) and property preprocessing, tokenization procedure, embedding based pretraining using Skip-gram and the training of an LSTM model based on the embeddings. The experiments section of the paper first provides a t_SNE based analysis of the learned embeddings showing some clustering and intermingling, followed by results on materials property prediction based on Materials Project. The paper provides an analysis of multiple regression and classification tasks and compares to relevant baselines for those tasks, including GNN-based methods and some embedding based methods. The final part of the experiments section describes using the CIFSemantics embeddings with an XGBoost model to try to better understand the utility of the embeddings.

**Strengths:**

The paper presents has the following strengths:
* A novel technique for text-based embeddings of materials systems directly from CIF files that is similar to word2vec. If successful, this technique could be used for a variety of use cases in materials science (originality).
* A new tokenization and processing approach for CIF files that could be useful for additional representation learning of materials systems (originality, significance)

**Weaknesses:**

While the paper presents an interesting idea, it has some major weaknesses in clarity and depth of the experimental results presented:
* The analysis is currently limited to one dataset (Materials Project), while other options exists - some of which the paper mentions in the related work section including OQMD [1], NOMAD [2].
* The paper does not describe related work in materials modeling such as the OpenCatalyst Dataset [3], the Open MatSci ML Toolkit [4] as well as work in materials science language modeling, such as MatBERT [5], MatSciBERT [6], MatSci-NLP [7]. This is relevant context that is missing in the current version.
* The experimental results are limited to just one dataset and one model architecture (LSTM). It would be good to have additional architectures (e.g., MLP, RNN, Transformer) in the analysis to understand the effects of those on performance.
* In order to understand the goodness of the proposed CIFSemantics embeddings, it would be good to have a comparison to other language model based embeddings including those from MatBERT, MatSciBERT and HoneyBee [8] as an example. This would give a better sense of how good the CIFSemantics are in comparison to other options.
* The tables are generally presented in a confusing way that is difficult to read, making it hard to understand the significance of the results.

[1] Kirklin, S., Saal, J.E., Meredig, B., Thompson, A., Doak, J.W., Aykol, M., Rühl, S. and Wolverton, C., 2015. The Open Quantum Materials Database (OQMD): assessing the accuracy of DFT formation energies. npj Computational Materials, 1(1), pp.1-15.

[2] Draxl, C. and Scheffler, M., 2019. The NOMAD laboratory: from data sharing to artificial intelligence. Journal of Physics: Materials, 2(3), p.036001.

[3] Chanussot, Lowik, et al. "Open catalyst 2020 (OC20) dataset and community challenges." Acs Catalysis 11.10 (2021): 6059-6072.

[4] Miret, Santiago, et al. "The open MatSci ML toolkit: A flexible framework for machine learning in materials science." Transaction on Machine Learning Research (2023).

[5] Trewartha, Amalie, et al. "Quantifying the advantage of domain-specific pre-training on named entity recognition tasks in materials science." Patterns 3.4 (2022).

[6] Gupta, Tanishq, et al. "MatSciBERT: A materials domain language model for text mining and information extraction." npj Computational Materials 8.1 (2022): 102.

[7] MatSci-NLP: Evaluating Scientific Language Models on Materials Science Language Tasks Using Text-to-Schema Modeling](https://aclanthology.org/2023.acl-long.201) (Song et al., ACL 2023).

[8] Song, Yu, et al. "HoneyBee: Progressive Instruction Finetuning of Large Language Models for Materials Science." arXiv preprint arXiv:2310.08511 (2023).

**Questions:**

* Is it possible to reorder the table to have methods as rows and properties as columns? This is more standard format for ML papers and would make things easier to read as all prediction results would be in a single column.
* Is there a reason you only performed your analysis on Materials Project when other sources of CIF files are available?
* What are types of model architectures can you think of that could benefit from the CIFSemantics embeddings? Are there creative ways to infuse them into modern NLP workflows?
* Are there other tokenization approaches that you considered in addition to the one outlined in the paper?

---

### Official Review · Reviewer_DZcp · 2023-11-02

**Soundness:** 2 fair
**Presentation:** 2 fair
**Contribution:** 2 fair
**Rating:** 3
**Confidence:** 5

**Summary:**

This paper proposes a novel NLP-inspired approach for representing and predicting properties of materials using crystallographic information files (CIFs). The key ideas are:

View CIFs as textual data, with atoms as "words" in context of their neighboring atoms and positions. This allows capturing local chemical environments and global crystallographic patterns.
Use Word2Vec on CIF corpus to learn vector embeddings for atoms. Atoms cluster based on periodic table trends, indicating embeddings capture intrinsic chemical characteristics.
Refine embeddings by predicting 15 diverse material properties with an LSTM model. Achieves accuracy comparable to state-of-the-art models tailored for specific properties/materials.
Demonstrate utility of embeddings for property prediction when only stoichiometry is known. Performs well for tasks like Curie temperature, diffusion barriers, stable compositions.
Proposed approach is generalizable, interpretable and requires no feature engineering. Represents a new direction for machine learning in materials science.

**Strengths:**

- Novel idea of treating CIFs as text for learning material representations is intuitive and impactful.
- Comprehensive evaluation across 15 properties and comparison to specialized models is impressive.
- Visualizations clearly show embeddings capture periodic table trends.
- Tests on stoichiometry-only data highlight adaptability of learned representations.
- Requires no hand-crafted features tailored for specific properties or materials.

**Weaknesses:**

- The whole presentation is clear but the idea is not quite good. The word2vec is an old algorithm, the authors are encouraged to use new algorithms, e.g, GPT. Look at this paper: https://arxiv.org/pdf/2307.04340.pdf
- The network is also out-of-date, why not Transformer?
- The paper looks like an initial report instead of a technical paper.
- Hyperparameter tuning and ablation studies could further optimize model performance.
- Quantitative analysis of learned embeddings could offer more insights.

**Questions:**

NA

---

### Official Review · Reviewer_DVQC · 2023-11-04

**Soundness:** 2 fair
**Presentation:** 3 good
**Contribution:** 2 fair
**Rating:** 3
**Confidence:** 5

**Summary:**

This paper introduces an approach to deciphering material properties using text-based method (LSTM) on crystallographic information files (CIFs). The study uses an unsupervised strategy that harnesses the underutilized potential of CIFs, producing atomic embeddings that capture intricate atomic relationships. The model, CIFSemantics, adeptly predicts 15 distinct material properties from the CIFs. The paper also discusses the conventional methods used in predicting material properties in condensed matter physics and materials science, and compares the performance of CIFSemantics to specialized models in predicting material properties.

**Strengths:**

- The idea is easy-to-follow and the writing is clear.

- This paper evaluates on 15 different properties that are relatively comprehensive for better understanding the capability and expressive power of this method.

**Weaknesses:**

- The novelty is limited. The skip-gram method for embedding words and the LSTM architectures are both out-of-dated on handling text data. I would expect this method propose a novel text-based learning technique on obtaining good materials representations from text.

- The compared baselines are inadequate. Some more recent and powerful methods [1, 2] have been proposed and should be reported against the proposed method empirically.

Overall, I would recommend this paper to resubmit to some domain-specific journals.

[1] Yan, Keqiang, et al. "Periodic graph transformers for crystal material property prediction." Advances in Neural Information Processing Systems 35 (2022): 15066-15080.

[2] Lin, Yuchao, et al. "Efficient Approximations of Complete Interatomic Potentials for Crystal Property Prediction." arXiv preprint arXiv:2306.10045 (2023).

**Questions:**

- How would LSTM compare to Transformers on learning materials literatures?

- Can we utilize text embedding directly from LLMs or any other types of pretrained languge models and then finetune upon them?

- On bandgap prediction, the model cannot beat CGCNN which is a 6-years old GNN-based model. Do you have more insights about why the text-based model is bad on such tasks?